# Catering Work Profession and Medico-Oral Health: A Study on 603 Subjects

**DOI:** 10.3390/healthcare9050582

**Published:** 2021-05-13

**Authors:** Sabina Saccomanno, Stefano Mummolo, Silvia Giancaspro, Rebecca Jewel Manenti, Rodolfo Francesco Mastrapasqua, Giuseppe Marzo, Vincenzo Quinzi

**Affiliations:** 1Department of Health, Life and Environmental Science, University of L’Aquila, Piazza Salvatore Tommasi, 67100 L’Aquila, Italy; stefano.mummolo@cc.univaq.it (S.M.); silvia.giancaspro@student.univaq.it (S.G.); rebeccajewel.manenti@student.univaq.it (R.J.M.); giuseppe.marzo@univaq.it (G.M.); vincenzo.quinzi@univaq.it (V.Q.); 2 ENT Department, Rivoli Hospital, ASL TO3 Torino, Italy; rodolfomastrapasqua@gmail.com

**Keywords:** oral health, catering workers, chef, professional illness, diabetes, cardiovascular diseases, obstructive sleep apnea syndrome, body mass index

## Abstract

The present observational prospective study analyzes the eating habits in association with the medico-oral health of catering workers, since they seem the category at higher risk of developing dental problems. Objective: To evaluate oral and medical health through a questionnaire in a total sample of 603 participants. Additionally, this article aims to provide information regarding the medico-oral implications of an unbalanced diet linked to this particular profession. Material and methods: A questionnaire created through Google Forms platform was sent to all members of the Italian Federation of Chefs (FIC). The mean age was 46.9 ± 32.6 ranging from 17 to 66 years old. Results: Catering workers’ years of service showed a significant impact on the presence of teeth pain (*p* < 0.05), missing teeth (excluding the third molars) (*p* < 0.01), treatment with prothesis (*p* < 0.01), dental fillings (*p* < 0.01), dental extractions (*p* < 0.01), diabetes (*p* < 0.05), high blood pressure (*p* < 0.01), joint pain (*p* < 0.01), back pain (*p* < 0.01), neck pain (*p* < 0.01), and gastroesophageal reflux (*p* < 0.05). Conclusions: A conspicuous part of the study sample was overweight, with a high BMI. Moreover, the years of service in this field showed significant impact on dental problems. Therefore, a very important role is that of preventing dental problems and giving information on the causes and effects unknown to many workers in the sector.

## 1. Introduction

Having a good oral health is important in maintaining a healthy mouth, teeth, and gums, and it is strictly correlated to a correct diet and a good general health [1].

In fact, an unbalanced and excessively caloric diet related to catering workers profession acts not only on oral health in the absence of a hygiene protocol but also on general health leading to obesity and, in some cases, to metabolic syndromes [2]. Obesity is defined as excessive body adiposity, above the ideal levels for good health. It develops from a chronic, positive energy balance, under the influence of multiple factors of social, behavioral, and environmental origin [3,4]. Obesity is assessed by body mass index (BMI). High BMI indexes are correlated with chronic diseases such as hypertension, dyslipidemia, type 2 diabetes [5], cardiovascular diseases (CVD) [5,6,7], respiratory disorders (e.g., obstructive sleep apnea syndromes (OSAS)) [8], and metabolic syndromes, as well as the development of some types of cancers [9]. The pathophysiology of obesity is multifactorial, involving in its development inadequate life style, neuronal, and hormonal mechanisms, as well as genetic and epigenetic factors [3,9]. The modified fatty acid composition of a Western diet, which is usually rich in saturated and trans-fatty acids, increases the risk of chronic vascular disease by elevating (blood serum) concentrations of both total and LDL cholesterol [9,10]. Moreover, diets high in sodium and low in potassium may lead to a variety of chronic illnesses, including hypertension and stroke [9]. Excessive food intake is a major contributor to obesity. Another large contributor to obesity is the lack of physical activity and sedentary behavior measured by screen time [11,12]. Respiratory disorders (e.g., obstructive sleep apnea) or sleep disturbance (e.g., sleep duration, insomnia, or excessive daytime sleepiness) have been identified as factors that influence obesity. Thus, the screening of symptoms of this particular disorder (e.g., daytime fatigue or snoring) is important in the catering workers profession [2]. In fact, the accumulation of visceral fat in obese patients is a key risk factor for OSAS [4,8,13]. The diagnosis and treatment of this disease is often neglected, either by the lack of knowledge of dentists and physicians or by non-adherence of patients to treatment. However, diagnosing and treating is of fundamental importance and involves multiple specialties cooperating with each other [14,15]. The treatments can range from weight loss to maxillomandibular advancement and palatal expansion [13,15].

Related to oral health, an unbalanced diet, typical of catering workers, affects the health and the integrity of our teeth [16,17]. The new World Health Organization (WHO) guideline advices to reduce free sugar consumption below 10% of the energy intake (10 E%) or even better below 5 E% of the diet [18]. In general, a diet that is beneficial to both general and dental health is one that is low in free sugars, saturated fat, and salts [19].

There is a correlation between the development of caries and the frequency of sugar intake although Bowen et al. [20] concluded that rather than the frequency of ingestion, the development of caries is strongly related to the time that sugars are available to microorganisms in the mouth. A higher frequency in the mouth means more demineralization and less remineralization. In fact, usually chefs taste their foods many times a day increasing the frequency of ingestion. The form of food consumption is relevant to the development of caries. In fact, the form of fermentable carbohydrates directly influences the duration of exposure and retention of food on teeth [16]. Consuming sugary food for an extended time or consuming long-lasting sources of sugar, such as lollipops or candies or even sticky food, can increase the time of demineralization. Acid environment can alter roughness [21], hardness [22], fluorescence intensity [23], and flexural properties [24] of restorative and prosthodontic frameworks.

It is known that certain foods can affect the development of dental plaque. In this regard, previous studies [25,26] showed that subjects who follow a diet rich in carbohydrates are more prone to develop gingivitis compared to those who follow a diet low in carbohydrates. That is why, in this study, the preferred type of food of catering workers was analyzed. Moreover, an increase in dietary sucrose has been associated with increased plaque build-up and evidence of gingivitis in humans [26].

Additionally, a correlation was found between patients with chronic periodontitis (CP) and undiagnosed diabetes due to the presence of HbA1c serum levels. In fact, patients with CP and undiagnosed diabetes presented significantly higher serum levels of HbA1c compared to periodontally healthy controls [27,28,29].

Some observational studies have demonstrated a direct and positive association between periodontitis and coronary heart disease, known also as ischemic heart disease (CAD), including myocardial infarction, stroke, and CVD [25,30,31]. Thus, it is interesting to analyze the percentage of diabetes and high blood pressure in catering workers.

### Aim of the Study

The motivation of choosing catering workers as the sample of this study is because they seem to be the categories at most risk for developing dental and medical problems. In fact, there is a link between medico-oral problems and an unbalanced diet typical of this specific profession [5]. Therefore, the authors’ aim can be summarized in this way:Increase awareness in this topic by highlighting the importance of improving the quality of life in this specific category by providing information regarding the effects of an unbalanced diet.To evaluate, through a questionnaire, the oral and medical health in 603 catering workers. Analyzing the oral care that catering workers have and if their years of service and lifestyle helped lead to an impaired oral health.

## 2. Materials and Methods

### 2.1. Study Context and Participants

This investigation is an observational prospective study, which involved a descriptive statistical analysis of a sample of 603 subjects, members of the Italian Federation Chefs (FIC) in Italy. The study was realized by the University of L’Aquila from January 2020 to February 2021. The present protocol was approved by the Ethics Committee of the University of L’Aquila (Document DR206/2013, 16 July 2013). The total number of all members of the FIC is 14,000, and 607 participants responded to the questionnaire. Four participants were excluded because they did not belong to the catering working field, making the final sample of 603 participants [32].

The inclusion criteria of the sample were enrolling subjects who are member of Italian Federation Chefs (FIC) and belonging to a catering worker category. The sample size was calculated with a 5% margin of error accepted, a confidence level of 95%, and a response distribution of 50%, and thus, the recommended sample size was 374.

### 2.2. Study Design and Construction of the Questionnaire

A detailed medical history was performed for all participants through a questionnaire. The conducted questionnaire was developed in Italian language.

A pre-pilot interview questionnaire was distributed and examined by six catering workers (2 chefs, 2 restaurant consultants, and 2 hotel management school students) aged between 18 and 50 years old to identify any ambiguities that it may contain.

### 2.3. Questionnaire Design

The questionnaire was built around open-ended questions and closed-ended questions (multiple choices). Google Forms was used to create the 54 questions. The time spent to answer the questionnaire varied from 7 to 10 min. The questionnaire gathered information on the medical history of hypertension, diabetes (including symptoms of nocturnal enuresis, gastroesophageal reflux, and dry mouth), daytime symptoms of OSAS (including sleepiness and headache), and nighttime symptoms (including habitual snoring). Drinking habits, dysgeusia, perceived stress scale, and self-control were also analyzed. Regarding oral care habits, data were collected related to frequency of home oral hygiene, use of mouthwashes, dental floss, and presence of bleeding while brushing. Data related to symptoms of temporomandibular disorders were taken into account including signs of backache, joint pain, bruxism, and temporomandibular joint disorders. Related to obesity, both suffocation episodes and the preferred food were analyzed.

### 2.4. Statistical Analysis

The mean value between the time since the last oral hygiene, dental visits, and dental X-ray was calculated by dividing patients into three groups: less than 12 months, between 12 and 24 months, more than 24 months, and thus, we performed a fisher exact test for dichotomic variables. Using Pearson correlation index, we confirmed a correlation between flossing, mouthwash, dental picks, and daily toothbrushing habit (*p* < 0.001). The same procedure was applied for questions regarding dental pathologies such as dental fillings, loss of teeth, and dental prosthesis. We performed a multinomial logistic regression for diabetes probability considering age, years of service, and BMI. For years of service and age, we performed multiple monovariate logistic regressions for dichotomic variables with Bonferroni correction.

## 3. Results

Employment of all participants is represented in Table 1.

Participants’ mean age was 46.9 ± 32.6 ranging from 17 to 66 years old. A conspicuous part of the study sample was overweight (mean 28.8 ± 5.1, ranging from 17.3 to 51.7 with a median of 27.6) (Table 2).

Data related to preferred foods evidenced that carbohydrates such as bread, pasta, or pizza were preferred by 518 participants (86.3%), while meat was cited by 180 (29.8%), vegetables were cited by 123 (20.1%), and “various” by 56 subjects (9.2%). Regarding professional oral hygiene, participants report that 9.9 ± 26.8 months have passed since the last professional hygiene, 9.6 ± 14.4 months since the last visit, and 21 ± 35.8 since the last X-ray. Related to home oral hygiene techniques, 151 people referred the use of dental picks, while 418 (69.3%) participants used mouthwash.

Regarding oral hygiene home habits, the majority of the sample brushed their teeth twice a day (49.1%), while 30.2% declared to brush their teeth three times a day, 20.4% brushed only once a day, and 0.3% of the sample stated that they do not brush their teeth every day. (Table 3).

Related to oral pathologies, data show that 56.4% of the sample reported to miss one or more teeth, excluding the third molar teeth mobility. Furthermore, 13.3% of the sample reported teeth mobility and 40.1% of the sample experienced gingival bleeding. Dry mouth was seen in 21.6% of the sample, while 17.1% of the participants’ evidenced alteration in mouth sensibility, the 5.1% mentioned taste alterations and the 31.2% declared to have gingivitis.

Regarding oral procedures (Table 4), the sample reported that orthodontic procedure and prosthesis treatment were common: 52.7% and 43.9%, respectively.

A table reporting the presence of the most relevant medical problems was realized (Table 5).

Another aspect analyzed in this study was the years of service. In fact, based on the questionnaire’s answers regarding the years of activity of catering workers, a correlation was seen with the presence of teeth pain (*p* < 0.05), missing teeth (excluding the third molars) (*p* < 0.01), treatment with prothesis (*p* < 0.01), dental fillings (*p* < 0.01), dental extractions (*p* < 0.01), diabetes (*p* < 0.05), high blood pressure (*p* < 0.01), joint pain (*p* < 0.01), back pain (*p* < 0.01), neck pain (*p* < 0.01), and gastroesophageal reflux (*p* < 0.05).

Moreover, years of service are correlated to snoring score (*p* < 0.05 coefficient 0.085) but not to bruxism, night awakenings, and daytime fatigue.

Considering diabetes, a binary regression testing BMI, years of service, and age was performed. The only significant aspect was the BMI value (*p* < 0.001 expected 1.18) regarding years of service since it showed a strong trend (*p* = 0.07 expected 5.43), while age was not considered significant (*p* = 0.4 expected 1.003).

## 4. Discussion

The aim of this article was to provide information regarding the medico-oral implications of an unbalanced diet linked to this particular profession. In fact, carbohydrates such as pizza and pasta resulted to be the favorite food in 86.3% of catering workers. In this study, it is possible to hypothesize a link between catering workers and high BMI since a conspicuous part of the sample was overweight.

Although work-related risk factors associated with CVD and T2DB have been well researched, there is no detailed knowledge regarding disparate occupational groups with a different risk exposition. Therefore, literature review was performed to investigate on this topic.

In this study, diabetes and high blood pressure seems to be relatively common in the sample population since 25.9% of the total individuals reported to suffer from them.

The oral condition and medical problems of catering workers were compared with other workers. In fact, a previous study [5] has demonstrated that chefs have a higher risk of CVD compared to office workers. A comparison of two occupational groups (45 chefs and 48 office workers) was studied with a focus on nutritional and psychosocial factors. The groups matched both for gender male with an average age of 30–45 years. According to Hartung et al. 2010, average chefs showed one risk factor more compared to office workers. The most frequent risk factors in both groups included overweight/obesity (chef group (CG): 62.2%; office group (OG): 58.3%) and elevated serum levels of total cholesterol (CG: 62.2%; OG: 43.8%). Even if, chefs often had higher CRP-concentration (40.0%), office workers suffered more from hypertension (37.5%). Instead, chefs showed a significant higher concentration of saturated fatty acids and oleic acid (OA) than office-workers. OA, the most prevalent omega-9-fatty acid, was found more significantly in the red blood cells of chefs. In our investigation, a questionnaire on medical health was submitted to the sample.

BMI of 30 kg/m^2^ or higher is used to identify individuals with obesity. A BMI of 25.0–29.9 kg/m^2^ is defined as overweight [33]. BMI in our sample is considered high, since a conspicuous part of the study sample was overweight (mean 28.8 ± 5.1, ranging from 17.3 to 51.7 with a median of 27.6). This aspect is directly linked with what is mentioned in the introduction section, since it is observed that overweight is one of the predisposing factors of OSAS, and it is a common aspect in catering workers. Diagnosis and treatment of OSAS is very relevant for the health of the population [13,15]. Dentists should detect signs and symptoms of obstructive sleep apnea syndrome. In our study, through the questionnaire submitted to the sample, it was possible to appreciate data related to the daytime and nighttime symptoms of OSAS. In fact, 31.1% of the sample demonstrated to have frequently sleepiness during the day, while 23% of the catering workers referred snoring while sleeping. It is understandable that a polysomnographic analysis should be appropriate in order to make an accurate diagnosis, but the importance of a questionnaire and of an accurate medical history was highlighted as a first step of the medical diagnosis. The treatment of choice is influenced by the etiology of the problem, but also by personal yearnings, such as the desire or not to change the facial appearance or the acceptance or not of sleeping with a CPAP mask beside the partner and the socio-economic characteristics of the patients [14,34]. Faber et al. 2019 [14] concluded that mandibular advancement devices (MADs) are a solid treatment option for primary snoring and mild or moderate OSAS. In fact, patients with severe apnea who are non-adherent to CPAP may also be treated with MADs. The study also underlined that the maxillomandibular advancement surgery is a safe and very effective treatment option to OSAS. According to Attali et al. 2019 [35], MAD revealed “latent dyspnea” related to the severity of upper airways mechanics abnormalities in OSAS patients. Similar findings were illustrated by Marty et al. 2017 [36] stating that custom-fitted MAD improved respiratory and somnolence parameters, with response rates similar to those published in the literature with other devices. Moreover, Quinzi et al. conducted a meta-analysis on 102 children affected by obstructive sleep apnea syndrome and concluded that on a mean follow-up duration of ≤3 years, a decrease of 66.1% of apnea-hypopnea index was detected in the sample after rapid maxillary expansion treatment [13].

In our investigation, various factors related to catering workers were analyzed in the questionnaire. Present findings suggest that self-reported bruxism and psychological states, such as anxiety or stress, may be related in working age subjects. In our study, 17.7% of the subjects stated to frequently suffer from bruxism [37]. The tension in the stomatognathic system caused by a stressful work environment or an incorrect posture [38] can also lead to TMJ pain, which is found in 32.7% of the subjects. A high percentage of the subjects suffered from back pain (76.6%) and diffuse joint pain (60.5%). Many studies reported a relationship between poor static posture and back pain. People with back pain should be aware of their static posture and develop good muscle strength and endurance in order to maintain a good performance at work [39,40]. Considering that head and trunk postures are among the factors affecting back pain and joint pain, correcting the working posture gains importance [41]. Another important data that were analyzed were the percentage of workers that suffered from neck pain (57.9%). Neck pain or neck dysfunction is a musculoskeletal disorder caused by improper posture with physical impairment or functional limitation. Above all, it is important to state that an incorrect position at work can cause changes in the alignment of the spine, leading to an improper posture and the appearance of pain [40,42].

Regular dental visits are important because they can both help prevent the onset of dental and oral diseases, and they can help diagnose certain diseases that have oral manifestations [43,44]. The frequency of dental visits depends on the oral health level of each individual and on several factors regarding general health, drug intake, and age [29]. Regarding oral treatments, obturations (81.6%) and extractions (81.6%) were very common treatments done to catering workers. The high frequency of obturation is an expected result, since there is a correlation with sugar intake and decay development. In fact, a link is suspected between the catering worker profession and dental decay. During their professional activity, they are encouraged to taste many times the food they are preparing, thus increasing the frequency of sugar intake and decreasing the remineralization time. Thus, their lifestyle may contribute to an imparted oral health. According to the American Dental Association’s general recommendations for the prevention of caries, an individual should brush their teeth twice a day with a fluoride containing toothpaste for 2–3 min, expressed also as at least 30 s per quadrant [45]. The fluoride content in the toothpaste should be of 1400–1500 ppm for adult population [29].

The prevention and control of dental caries and periodontal diseases and the prevention of ultimate tooth loss is a lifelong commitment [25,46]. According to literature, the commonly utilized spices to stimulate the taste buds, whole or powdered chili, have an inevitable importance. The active component of the spice, Capsaicin possesses the antioxidant, anti-mutagenic, anti-carcinogenic, and immunosuppressive activities having ability to inhibit bacterial growth and platelet aggregation [47].

As a general recommendation, a patient should perform a professional dental cleaning at least every 6–8 months to prevent the evolution of periodontal disease and caries formation. Additionally, dental implication related to eating habits should be mentioned. In fact, chemical factors (acids), biological factors (amount and composition of saliva and its buffer system), and habitual factors (dietary and hygienic habits) may cause erosive lesions due to an imbalance between protective and damaging factors [48].

This work is original and innovative but presents a critical issue from the bibliographic research on PubMed since there are a very few or too old studies done on this topic. Above all, there are no studies related to specific professional categories (Chef—oral health) exposed to this risk. Many articles, however, highlight that the exposure to an etiological factor for a long time, as it happens in work activities, is responsible for the onset of diseases.

## 5. Limitations

The risk of bias in the sample population’s answers could be present even if there was control over who actually completed the questionnaire since it was sent only to the members of the FIC. A possibility of error, derived by the inevitable subjectivity on the catering workers answers, has to be considered. In addition, the COVID-19 pandemic started to take over and other projects we had related with this study could not be finalized. We hope in fact that as soon as we are able to go back to having public events related to this category, we can undergo further investigations in this sector.

## 6. Conclusions

Based on the results, BMI had a high value in the entire sample, and it does not depend on the age or on the years of activity. A prevalence of oral health problems as well as general health disorders was observed among catering workers. In fact, years of service showed a significant impact on oral and medical problems.

Thus, prevention of oral health and the information regarding the causes and the effects of an unbalanced diet are unknown to many workers in this sector.

However, a larger sample study, analyzing also gender differences, could support and confirm the results of this work.

A future analysis could be appropriate to strengthen the results found in this investigation.

## Figures and Tables

**Table 1 healthcare-09-00582-t001:** Occupation of the sample (N = 603).

Employment	Percentage of Sample
Chef	84.25%
Catering workers	3.81%
Pastry chef	1.16%
Students	1.16%
Surchef	0.33%
Teachers	8.96%
Waiters	0.33%

**Table 2 healthcare-09-00582-t002:** Demographic data of the sample.

Years of Practice	Absolute	Proportion
0–5 (years)	33	5.5
6–1	77	12.8
12–17	81	13.4
More than 18	412	68.3
Age (years)	46.9 ± 32.6	
Height (cm)	174.7 ± 7.8	
Weight (kg)	86.2 ± 17.8	
BMI	28.8 ± 5.1	

**Table 3 healthcare-09-00582-t003:** Oral hygiene home habits.

How Many Times a Day You Brush Your Teeth?	Absolute	Proportion
0	2	0.3
1	123	20.4
2	296	49.1
3 or more	182	30.2
Mechanical or Electrical Toothbrush		
Electrical	192	31.8
Mechanical	411	68.2
Total	603	

**Table 4 healthcare-09-00582-t004:** Oral procedures.

Extractions	Absolute	Proportion
No	103	17.1
Yes	492	81.6
Does not know	8	1.3
Obturations		
No	94	15.6
Yes	492	81.6
Does not know	17	2.8

**Table 5 healthcare-09-00582-t005:** Presence of medical problems in the sample (*n* = 603).

	Absolute	Proportion
Temporomandibular joint pain	197	32.7
Neck pain	349	57.9
Back pain	462	76.6
Diffuse joint pain	365	60.5
Daytime Fatigue (Rate from 0 to 4)		
Never	14	2.3
Not frequently	81	13.4
Seldom	294	48.8
Frequently	189	31.3
Constantly	25	4.1
Snoring (Rate from 0 to 4)		
Never	43	7.1
Not frequently	81	13.4
Seldom	234	38.8
Frequently	157	23.0
Constantly	88	14.6
Clenching and/or bruxing		
Never	205	34.0
Not frequently	107	17.7
Seldom	140	23.2
Frequently	107	17.7
Constantly	44	7.3
Nocturnal enuresis		
Never	44	7.3
Not frequently	144	23.9
Seldom	210	34.8
Frequently	128	21.2
Constantly	77	12.8
High blood pressure	156	25.9
Diabetes	22	0.4
Gastroesophageal reflux	222	36.8
Perceived Stress Scale		
Mean	9.1 ± 3.9	
Median	9	

## Data Availability

The data that support the findings of this study are available from the corresponding author (S.S.) upon reasonable request.

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
