# Peer review of "Catering Work Profession and Medico-Oral Health: A Study on 603 Subjects"

_healthcare, 2021, doi:10.3390/healthcare9050582_

Round 1

Reviewer 1 Report

Dear Authors,

the observational study “Catering work profession and medico-oral health: a study on 603 subjects” aims to report the information about eating habits in association with medico-oral health of catering workers obtained from an administered questionnaire to improve the quality of the life.

The topic resulted interesting, but some parts, in my opinion, must be modified.

In the abstract and also in the results, the Authors report a mean age of 46.9 + 32.6. The obtained values ​​of the mean plus or minus standard deviation, result not into the reported range (from 17 to 66). Furthermore, please add the standard deviation preceded by the sign plus-minus, in the abstract.

At the first impact, the association between the goals described in the aims and the topics reported in the manuscript result not clear. In particular, in the introduction, the theme is oriented mainly to obesity. Moreover, the motivation of choosing the category of catering workers discussed in the following steps of the article is not well defined. The Authors, in fact, cite the catering workers only in the aim without contextualized the problem of this worker's category in the above paragraphs.

My suggestion is to the reorganized introduction, highlighting the motivated choosing of the population, and to reorganize the aims in defined points.

Figure 1 in my opinion should be modified because if the article results printed in white/black, the reader is not able to recognize the categories. So, I suggest modifying the color with motif, or to create a table with the data. In case of the graph will be maintained, please enlarge the label containing the data.

In table 4, there is a “never” in bold. Furthermore, I suggest slightly separate the reported medical problems to facilitate the reading.

Furthermore, the discussion should be completed in accordance with introduction reorganization.

Author Response

Good morning,

Thank you for considering our article.

I am attaching the response to the reviewer's comments 

Best,

Sabina Saccomanno

Reviewer 2 Report

This study examined the relationship between oral and medical health in catering workers. I think this study is important in clarifying occupational illness. However, I think there are some points need to be revised before publication.

Figure 1 and Table 1 should be described at "Results" section.

L119-121: "The sample size was calculated...the recommended sample size was 374." I think this sample size calculation is unclear. It is necessary to estimate the effect size from previous studies.

L168-169: This sentence is different from the content in Table 2.

L188: catering workers' years?

Table 4: Diabetes (N=22, 25.9%) Is it true?

In the Discussion section, the authors should consider the characteristics of occupational illness of catering workers. I think the oral condition and medical problems of catering workers should be compared with other workers by referring to the other research reports. 

Author Response

(The authors gave the same response as above.)

Round 2

Reviewer 1 Report

Dear Authors,

thank you for your collaboration.

I am satisfied with the applied revision.

I have only a minor relatively table 5, so to vertically center the items in each line, to help the reader.

Best Regards

Author Response

Thank you for your generous collaboration and your precious comments. 

We have modified table 5 according to your helpful suggestion. 

Best Regards, 

Sabina Saccomanno

Reviewer 2 Report

Thank you for your answers to my comments and revision of the manuscript. I have no more comments for the revised manuscript.

Author Response

Thank you for your valuable and precious comments. 

Best regards, 

Sabina Saccomanno

This manuscript is a resubmission of an earlier submission. The following is a list of the peer review reports and author responses from that submission.